# Multimodal Elastography of the Main Salivary Glands—A Narrative Review

**DOI:** 10.3390/diagnostics15040411

**Published:** 2025-02-08

**Authors:** Delia Doris Donci, Lavinia Manuela Lenghel, Cristian Dinu, Sebastian Stoia, Maria Bădărînză, Rareș Mocan, Carolina Solomon, Anca Ciurea

**Affiliations:** 1Department of Radiology, “Iuliu Hațieganu” University of Medicine and Pharmacy, 400012 Cluj-Napoca, Romania; 2Department of Oromaxillofacial Surgery, “Iuliu Hațieganu” University of Medicine and Pharmacy, 400012 Cluj-Napoca, Romania; 3Department of Rheumatology, “Iuliu Hațieganu” University of Medicine and Pharmacy, 400012 Cluj-Napoca, Romania

**Keywords:** salivary glands pathology, parotid gland, submandibular gland, strain elastography, shear wave elastography, viscoelastography

## Abstract

Elastography has emerged as a valuable imaging technique that evaluates tissue stiffness and offers complementary insights into conventional ultrasonography. The aim of this article is to review the utility of elastography in assessing salivary gland pathologies. The review categorizes findings by pathology and the physical principles underlying each elastographic modality. Key modalities discussed include strain elastography, shear wave elastography, and novel hybrid techniques, such as viscoelastography, highlighting their strengths, limitations, and clinical applications in salivary gland imaging.

## 1. Introduction

Salivary glands, as an organ system, encompass one of the widest spectrums of pathology, ranging from developmental anomalies and inflammatory disorders (acute, caused by infections or lithiasis; and chronic, caused by autoimmune diseases or following radiotherapy) to various benign and malignant neoplasms, posing significant diagnostic and therapeutical challenges.

Due to their superficial location, salivary glands are easily accessible with high-resolution ultrasonography (US), which represents the initially performed imaging technique when clinically indicated. Elastography assesses tissue stiffness and provides complementary information to conventional US, being regarded as a valuable diagnostic tool, especially in evaluating diffuse salivary gland pathologies [1]. Elastography is mainly divided into strain elastography and shear wave elastography, following the assumption that the examined structures present a simple behavior and are elastic, linearly uniform, and isotropic. However, biological soft tissues are naturally characterized by two mechanical properties: elasticity (response to deformation) and viscosity (reaction to deformation rate). Both properties influence the shear wave propagation process. Therefore, a novel imaging technique has emerged, viscoelastography, which assesses both tissue elasticity and viscosity, the former linked to shear wave speed and the latter linked to shear wave dispersion [2,3]. Viscoelastography is largely unexplored, given its limited availability in the current US device systems.

This article aims to provide a review of the utility of elastography in assessing major salivary glands pathology. In this review, we conducted a computerized search using the PubMed database (www.ncbi.nlm.nih.gov/pubmed/, accessed on 30 August 2024). The following search terms were utilized: “salivary glands elastography”, “parotid gland elastography”, “submandibular gland elastography”, and “viscoelastography”. Only English-language articles were included. The titles and abstracts of the search results were assessed, and a total of 110 articles were deemed appropriate for full-text analysis.

The articles were reviewed and discussed in relation to the pathology and the fundamental physical principles of the elastographic method. The use of elastography was addressed in the study of normal glands, diffuse salivary gland diseases (such as primary Sjögren’s Syndrome, radiation therapy-induced injuries, and sialolithiasis), and salivary gland tumors.

## 2. Elastography Principles and Methods

Elastography represents a noninvasive imaging technique complementary to ultrasonography that provides information regarding tissue stiffness in vivo. Stiffness reflects the tissue’s resistance to deformation when subjected to an applied external force. Generally, all elastographic methods include mechanical excitation and subsequent monitoring of the resulting tissue reaction [2,4,5].

Based on the measured physical quantity, elastography is mainly divided into strain elastography (SE) and shear wave elastography (SWE). Therefore, stiffness images are generated either as a result of a tissue strain or deformation (SE) or by measuring shear wave propagation speed inside tissues (SWE) [2].

A classification overview of the elastographic methods is offered in Figure 1. Both SE and SWE follow assumptions that tissues have a simple behavior (constant density, linear, homogenous, and isotropic).

### 2.1. Strain Elastography (SE)

SE is a technique that can generate tissue strain by using mechanical excitation through manual compression or physiological cardiovascular/respiratory pulsations. SE can also generate tissue displacement by using an acoustic radiation force impulse, typically generated by a focused ultrasound beam. Strain is calculated by assessing tissue displacement in the direction of pulse propagation [6].

The main display methods for SE are:Qualitative parameters:
Elastogram = strain image in which the normalized strain is used to represent the average value within a region of interest (ROI) to provide a stable image that is unaffected by variations in compression intensity. Elastograms are color-coded maps that can be dynamically displayed alongside or superimposed on gray-scale images. Depending on the lesions’ stiffness pattern in comparison to the surrounding tissues, several elastographic scores have been proposed [2,7].
Semiquantitative parameters:
Strain Ratio = ratio between the lesion’s strain and the reference tissue’s strain [8].E/B size ratio = ratio between the lesion’s size in the elastogram (E) and its size in the ultrasound B-mode image (B) [9].

### 2.2. Shear Wave Elastography (SWE)

The SWE technique generates shear waves within a tissue by using either an acoustic radiation force impulse (ARFI) or external vibrations, as an excitation. Shear waves propagate perpendicularly to the direction of the pulse/vibration and tissue displacement [2,10].

In SWE, tissue stiffness can be quantitatively determined by measuring shear wave velocity in meters/seconds (m/s), which can be converted to Young’s modulus in kilopascals (kPa) using the following formula [2]: E = 3ρc_s_^2^, where: E = Young’s modulus; ρ = tissue density; c_s_ = propagation speed.

SWE techniques include transient elastography (TE), which uses controlled body-surface vibrations to generate shear waves, which then propagate to the organ of interest. TE is embodied on a dedicated device (Fibroscan^®^) used only to quantitatively assess liver stiffness in kPa, without displaying an anatomical image [11].

Other SWE techniques rely on ARFI to generate shear waves directly in the tissue and include point shear wave elastography (pSWE) and multidimensional SWE (2D-SWE, 3D-SWE). A linear or convex transducer emits focused ultrasound pulses (referred to as push pulses or ARFI) to produce shear waves. These pulses are sent out repeatedly within a brief timeframe and generate shear waves that propagate much slower than ultrasound waves. To determine the shear wave propagation velocity, B-mode tracking pulses are employed, which involves measuring the arrival time delay between two points located at a known distance one from each other [2,12]. pSWE assesses tissue stiffness within a focal (~1 cm^2^) region of interest, expressed only quantitatively (m/s or kPa). Multidimensional SWE (2D/3D) assesses tissue stiffness over a broader area, both quantitatively (m/s or kPa) and qualitatively, as a color-coded elastogram [12].

### 2.3. Viscoelastography—“The New Actor on Stage”

Viscoelastography represents a novel imaging technique that assesses both tissue elasticity and viscosity. Until recently, SE and SWE techniques followed the simplified assumption that the examined structures are elastic, linearly uniform, and isotropic [2,12]. However, biological soft tissues are naturally viscoelastic, and not purely elastic [4]. Therefore, the propagation of shear waves is influenced by both mechanical tissue properties: elasticity, which is linked to shear wave speed, and viscosity, which is linked to shear wave dispersion [3,4].

To overcome this limitation of most current elastographic techniques, which do not consider the dispersion effect, to the best of our knowledge there are currently two manufacturers (Supersonic Imagine and Canon) that have developed new imaging methods that include tissue viscosity properties in their algorithm. Supersonic Imagine has developed Viscosity Plane-wave UltraSound (Vi.PLUS) 2D imaging mode, embedded in the Aixplorer MACH 30 system, which allows visualization and quantification of tissue viscosity over a region of interest, expressed in pascal seconds (Pa.s). Canon has developed shear wave dispersion imaging (SWD) available on the Aplio i900 series diagnostic ultrasound system that provides a quantitative assessment of the dispersion slope (expressed in m/s/kHz), a parameter directly linked to viscosity. Dispersion is linked to the frequency dependency of speed and attenuation of the shear waves in the viscous component [13,14].

## 3. Clinical Applications of Ultrasound Elastography in Salivary Gland Pathology

Ultrasonography (US) serves as the primary imaging modality for the major salivary glands (MSGs) when clinically indicated, as they are easily accessible due to their superficial location [10]. Inflammatory and neoplastic changes in soft tissues are linked to alterations in their elasticity. Consequently, elasticity imaging provides complementary information to conventional US and is regarded as a valuable diagnostic tool in distinguishing between different pathological conditions [1].

### 3.1. Diffuse Salivary Gland Diseases

#### 3.1.1. Normal Values of Salivary Gland Stiffness

The normal stiffness values as determined by 2D-SWE (Figure 2) range within 5.46–9 kPa for the parotid gland (PG) and 8.63–11 kPa for the submandibular gland (SMG). Age, gender, and body mass index were not found to be significant confounding factors to the elasticity modulus of MSG [15,16]. No statistically significant differences were detected between the SWE velocity values of the MSG of smoker and non-smoker healthy subjects [17].

#### 3.1.2. Primary Sjögren’s Syndrome

Sjögren’s Syndrome is a chronic inflammatory autoimmune disease that primarily involves the salivary and lacrimal glands, leading to progressive xerostomia and keratoconjunctivitis sicca. Primary Sjögren’s Syndrome (pSS) can be diagnosed when there is no concurrent rheumatic disorder, while secondary Sjögren’s Syndrome is linked to the presence of another rheumatic condition, such as systemic lupus erythematosus, systemic sclerosis, or rheumatoid arthritis [18].

Generally, studies that evaluate the diagnostic performance of elastography in diagnosing pSS reveal a lower parenchymal gland elasticity during the disease, especially of the PG [19,20,21].

Several studies assessed the role of SE in evaluating pSS with promising results. PG and SMG presented higher elasticity scores, equivalent to increased stiffness, in patients with pSS compared to the non-pSS control group. Furthermore, the sum of the scores for all four salivary glands performed better in detecting pSS changes (AUC = 0.916) than the sum of the bilateral PG (AUC = 0.857) or bilateral SMG (AUC = 0.783). For the optimal cut-off score value of 9 in the combined assessment of all four salivary glands, the reported sensitivity and specificity were 81% and 87%, respectively [7].

Significant differences were also observed between strain ratios of MSG of patients with pSS and controls. Strain ratios, which are semiquantitative parameters, provided high sensitivity and specificity in diagnosing pSS for the cut-off value of 2.45 for PG (83% and 92%, respectively) and 1.55 for SMG (83% and 88%, respectively) [19].

In a study focused on chronic inflammatory disorders that included patients mainly with pSS, but also chronic recurrent parotitis and sialolithiasis, the strain ratios also proved to be higher in the major salivary glands in comparison to healthy controls. There were no significant variations in the median strain ratio values across cases with different causes of inflammation. This might be explained by the fact that chronic inflammation, regardless of the cause, triggers gradual destruction of the glandular acini, accompanied by lymphocytic infiltration, fibrous tissue replacement, and sialectasis, generally resulting in increased stiffness [20].

The role of shear wave elastography techniques has also been largely studied in patients with pSS. A study conducted by Arslan et al. involving 53 pSS subjects and 35 controls proved that between the two groups, there are significant differences in the shear wave velocities in the PG (3.1 ± 0.8 vs. 2.1 ± 0.3 m/s, *p* < 0.001) and SMG (2.9 ± 0.4 vs. 2.3 ± 0.2 m/s, *p* < 0.001). For the cut-off values of 2.48 m/s for PG and 2.59 m/s for SMG, the sensitivity and specificity in diagnosing pSS were 82.1% and 91.7% for PG, respectively, and slightly lower for SMG, 79.2% and 90.0%, respectively [21].

SWE also proved useful in assessing patients with secondary Sjögren’s Syndrome, revealing significantly higher SWV values in comparison to the control group [22].

Notably, a recent study has emphasized the effectiveness of employing the 2D-SWE technique in a subgroup of patients who presented either normal (Grade 0) or inconclusive (Grade 1) features during the ultrasound B-mode MSG evaluation, which allowed the diagnosis of pSS with 94% sensitivity for the stiffness cut-off value of 6.45 kPa. [21]. Furthermore, within this same study, which included eight patients diagnosed with pSS-related MALT lymphoma, the authors noted that the combination of Grade 3 findings in ultrasonography (indicating > 50% hyperechoic bands), along with parotid hypertrophy and elasticity values exceeding 11.5 kPa, exhibited 100% specificity for MALT lymphoma, while maintaining a high level of sensitivity (92%) [23]. Figure 3 shows increased SWE values in the parotid gland parenchyma of a patient with pSS and lymphoma versus a patient with pSS without lymphomatous proliferation.

Another study evaluated the ability of combined SWE with pixel analysis to diagnose pSS. A lower quantity of red pixels (representative of soft tissue) was discovered in the elastograms of patients with pSS. Less than 54% of red pixels, as a cut-off point risk for pSS, presented an OR of 3.8 [24].

In a recent meta-analysis including fifteen articles and a total number of 816 patients diagnosed with pSS and 735 healthy/disease controls, SWE presented high pooled sensitivity and specificity in differentiating between the two groups (0.80 with 95%CI: 0.71–0.87 and 0.87 with 95%CI: 0.78–0.92, respectively) [25].

Elevated stiffness parameters within the glandular parenchyma have been observed in individuals diagnosed with pSS in contrast to those with sicca symptoms that did not fulfill the diagnostic criteria for pSS. This suggests a possible application of elastrography in predicting the early onset of Sjögren’s Syndrome [26,27].

#### 3.1.3. Radiation Therapy-Induced Injuries

Radiotherapy represents one of the main treatment methods for head and neck malignancies. Radiation-induced salivary gland injuries, with subsequent xerostomia and speech difficulties, are widely prevalent and develop in up to 90% of cases [28,29].

Patients irradiated for head and neck squamous cell carcinoma presented increased stiffness of MSG assessed with 2D-SWE in comparison to healthy controls. No significant differences were noticed for patients who underwent conventional radiotherapy versus intensity-modulated radiotherapy. However, the effect of time since RT on elasticity values remains uncertain. The mean elastography values of MSG presented an ascending trend in the short-term follow-up, with values of 34.2 kPa, 36.6 kPa, and 46.0 kPa, obtained immediately after RT, and at 6 and 12 months, respectively. At 24 months follow-up, the values become lower (40 kPa), with no statistically significant difference. The investigators acknowledge that this trend requires further examination [30].

Another 2D-SWE study also revealed higher shear wave speed after radiation therapy in both PG (2.43 m/s vs. 1.99 m/s) and SMG (2.50 m/s vs. 2.32. m/s), with an associated significant decrease in size of the SMG and not of the PG [31].

Shear wave velocity assessed with p-SWE also showed higher values in MSG patients who underwent radiotherapy in comparison to normal glands [32]. Furthermore, pSWE proved useful in monitoring the impact of local liposomal salivary replacement on glandular stiffness in patients with prior radiotherapy and revealed how PG stiffness values improved after two months of therapy [29].

#### 3.1.4. Sialolithiasis

PG and SMG with evidence of calculi (sialolithisis) assessed with pSWE presented significantly higher shear wave velocity values in comparison to healthy contralateral glands in the same subject (4.19 m/s vs. 2.42 m/s in PG; 2.91 m/s vs. 2.16 m/w in SMG) [33].

One study proved that following successful treatment with interventional sialendoscopy surgery, the shear wave velocity of the affected gland with sialolithiasis significantly decreased. However, there still remained a significant difference in stiffness between the affected gland and the contralateral normal gland even 1.55 months after surgery [34].

### 3.2. Salivary Glands Tumors

There are contradictory results in studies that assess the diagnostic performance of SE in salivary gland tumors. A four-pattern scoring system to assess salivary gland tumors was proposed: tumors that are completely green, mostly green with some blue areas, mostly blue with some green areas, and entirely blue, corresponding to scores of 1, 2, 3, and 4, respectively. At a cut-off score ≥3, malignancy was detected with sensitivity, specificity, and accuracy of 54.5%, 56.4%, and 56.2%, respectively, by conventional SE, and 77.3%, 63.8%, and 65.4%, respectively, by ARFI-SE. Different SE patterns are presented in Figure 4. The stiffness of malignant tumors located in the deep parotid lobe was hard to assess with ARFI-SE, due to the attenuation of the acoustic push pulses [35].

Another SE study that also used a four-score system to assess salivary gland tumors (with scores of 3 and 4 attributed to malignancy), revealed 100% sensitivity and 100% negative predictive value, at a cost of low specificity (66%) and positive predictive value (52%). Fifteen out of forty-four benign lesions presented scores of 3 and 4. Therefore, the authors concluded that elastography alone cannot be used to discriminate malignant from benign salivary gland lesions [36].

Altinbas et al. assessed the role of the elasticity score (E-index) in assessing parotid gland masses. E-index provides an absolute value ranging from 0 (softest) to 6 (hardest). Malignant PGT presented higher elasticity scores than benign PGT (3.44 ± 0.85 vs. 2.75 ± 0.95, *p* = 0.034). However, the authors also report a significant overlap of stiffness scores between the two studied groups [37].

Conversely, one study reported a high diagnostic performance in differentiating between benign and malignant PGT using strain ratios. Using a cut-off value of 2.1, the reported sensitivity, specificity, and accuracy were 83.3%, 97%, and 94%, respectively. However, the study only included 39 lesions, with most of them of benign histology (84.6%), which might have impacted the obtained result [38].

Several papers have focused on the ability of SWE to distinguish between salivary gland masses. Studies suggest that malignant salivary tumors usually present a higher mean stiffness in comparison to benign tumors, however, there is a significant overlap of the reported cutoff values [39]. Among benign PGT, Warthin’s tumor is generally soft, but pleomorphic adenoma presents variable stiffness values, which fall in the range of malignant PGT [40,41].

Su et al. developed a nomogram including clinical, conventional ultrasonography, and SWE features, which demonstrated a good diagnostic performance in distinguishing pleomorphic adenoma from Warthin’s tumor, achieving AUC values 0.947 in the training cohort and 0.903 in the validation cohort. Decision curve analysis confirmed that the nomogram model surpassed the other studied models (clinical + conventional ultrasonography model; SWE model) in clinical utility [42].

SWE also proved to be particularly useful in enhancing the effectiveness of obtaining diagnostic tissue when employed for the guidance of fine needle aspiration cytology (FNAC) procedures. SWE-guided FNAC presented a lower incidence of false-negative results and non-diagnostic samples in comparison to B-mode US FNAC [43].

However, one meta-analysis that included ten eligible studies and 725 parotid tumors in total, concluded that overall, sonoelastography provides limited utility in distinguishing between malignant and benign parotid lesions. The pooled sensitivity and specificity were 0.67 and 0.64, respectively. This meta-analysis was heterogeneous and included several elastography techniques. Consequently, the performance of different elastography parameters was analyzed in each study, such as shear wave velocity, coefficient of stiffness variability, consensus sonoelastography scores, elastographic scoring, or elasticity contrast index. Semiquantitative and quantitative parameters achieved superior diagnostic performance in comparison to qualitative parameters, as the former are automatically generated by the ultrasound device and less reliant on the operator’s skills and expertise [44].

### 3.3. Viscoelastography—What Do We Know So Far?

Viscosity is a new imaging parameter linked to inflammatory tissue changes and is thought likely to have a significant influence on future clinical elastography diagnosis [45,46].

To the best of our knowledge, only two studies published in the literature have so far assessed the viscosity of the salivary glands using the Supersonic Vi.PLUS module. Normal viscosity values for the main salivary glands were proposed. The mean normal viscosity value for the PG was 2.13 ± 0.23 Pa·s, significantly lower than the mean viscosity value of the SMG of 2.44 ± 0.35 Pa·s [47]. An example of viscosity evaluation of the PG and SMG is presented in Figure 5.

Supersonic Vi.PLUS together with 2D-SWE.PLUS also proved to be a valuable imaging tool in functionally assessing the major salivary glands in healthy subjects. Viscosity and stiffness of parotid glands significantly increased following stimulation with a sialogogue agent, and no significant changes in these two parameters were observed for the submandibular glands. Gender and BMI were not confounding factors for these two parameters. Vi.PLUS measurements presented a good positive correlation with 2D-SWE.PLUS values for PG and SMG, before and after stimulation. Depending on the disease (e.g., inflammation or fibrosis), further studies are necessary to assess whether these correlations decrease or no longer exist [48].

### 3.4. Specific Challenges of Salivary Gland Elastography

The accuracy of the elastographic measurements can be influenced by artifacts that may occur in the neck region, often caused by the proximity to the skin and bony structures (ramus of the mandible and mastoid processes), or eventual focal protuberances of the skin. These factors create local heterogeneity and can distort the actual tissue elasticity [10].

Furthermore, the complexity of the structures encountered in the neck region poses a challenge in establishing an adequate ROI for the assessment of the elastogram, which is as a qualitative parameter highly dependent on the operator’s expertise [1].

An increased pre-compression with the ultrasound probe proved to induce a statistically significant rise in SWE values for both salivary glands. Therefore, to enhance the reliability and reproducibility of SWE in assessing salivary glands, the degree of pre-compression by the ultrasound transducer should be standardized [17].

Other elastography challenges regard salivary tumors with marked hypoechogenicity, which in turn leads to insufficient acoustic backscatters able to deliver reliable elasticity values. Bulging tumors cause difficulties in applying a linear transducer evenly onto the skin surface without inducing variations to the applied stress field, consequently leading to falsely elevated stiffness measurements [1].

## 4. Conclusions

In summary, elastography presents various clinical applications in assessing salivary glands, proving to be particularly useful in diffuse pathologies. However, the need for standardized methods and equipment calls for further research studies.

## Figures and Tables

**Figure 1 diagnostics-15-00411-f001:**
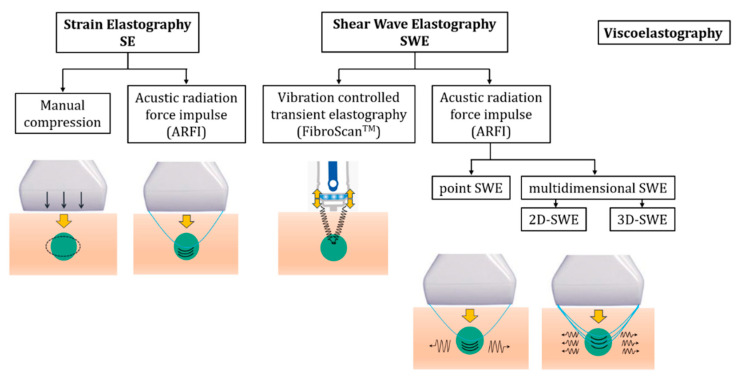
Classification of the elastography techniques; adapted after [2].

**Figure 2 diagnostics-15-00411-f002:**
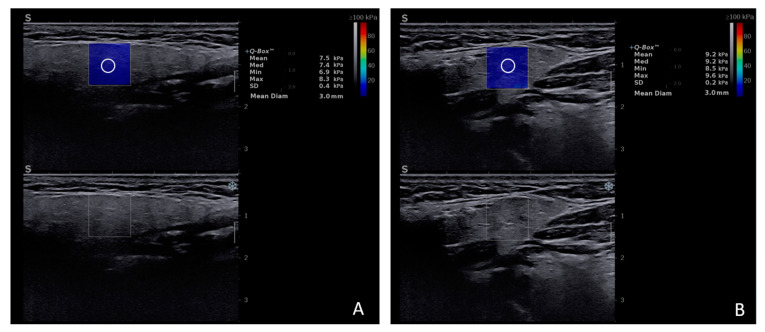
2D-SWE examination of the parotid gland (**A**) and submandibular gland (**B**) healthy subject.

**Figure 3 diagnostics-15-00411-f003:**
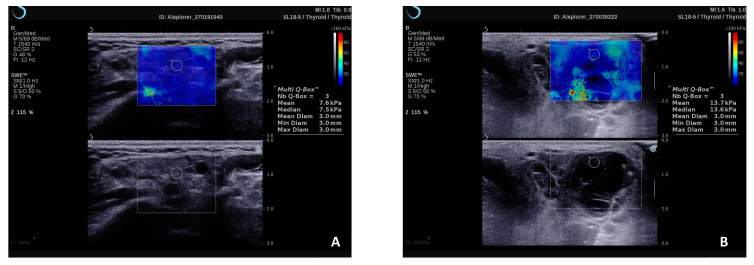
2D-SWE examination of the parotid gland in a patient with pSS without lymphomatous proliferation (**A**) and a patient with pSS and associated MALT lymphoma that demonstrates higher stiffness values (**B**).

**Figure 4 diagnostics-15-00411-f004:**
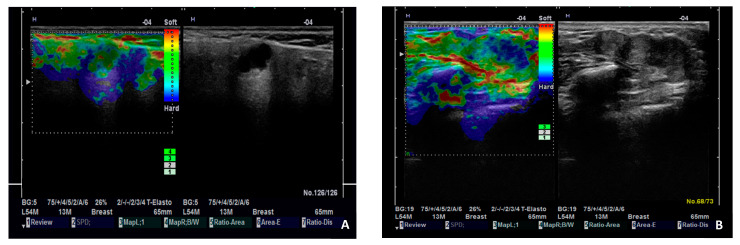
Illustration of SE patterns in parotid gland tumors. (**A**). Pleomorphic adenoma with SE score 1 (almost completely green). (**B**). Adenoid cystic carcinoma with SE score 3 (mostly blue with some green areas).

**Figure 5 diagnostics-15-00411-f005:**
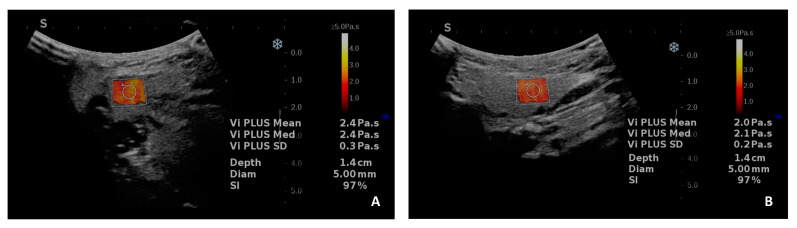
Viscosity assessment of the main salivary glands ((**A**). submandibular gland; (**B**). parotid gland) in a healthy subject using the Vi.PLUS module available on the Supersonic Imagine Aixplorer ultrasound device. A color-coded map is displayed in the Vi.PLUS box: high viscosity is represented by the colors white–yellow, while low viscosity is represented in red. Numeric viscosity results (the mean, median, and standard deviation values expressed in Pa.s) are displayed on the right side of the image.

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
