# Peer review of "Multimodal Elastography of the Main Salivary Glands—A Narrative Review"

_diagnostics, 2025, doi:10.3390/diagnostics15040411_

Round 1
Reviewer 1 Report
Comments and Suggestions for Authors
Thank you for the opportunity to review “Multimodal Elastography of the Main Salivary Glands - Narrative Review” by Donci et al. These investigators identify their goal to provide a review of the utility of elastography and assessing major salivary gland pathology through review of 110 articles access through pub Med
I enjoyed reading this review article and, as a clinician, found insights useful to my practice. There is value in presenting an update to salivary elastography in an understandable manner directed to physicians who are not also physicists.
Comments are enumerated below:
1. Attention to improving the grammatical presentation is warranted and is not limited to this example:
Elastography assesses tissue stiffness and provides complementary information to conventional US, being regarded as a valuable diagnostic tool, especially in the assessment of diffuse salivary gland pathologies [1].
2. As a review article, this work should clearly describe terminology likely to be foreign to the clinicians reading it (the first time those terms are used) including the terms elasticity and viscosity presented in a vague fashion the following segment
However, biological soft tissues are naturally characterized by two mechanical properties, elasticity, and viscosity, both influencing the shear wave propagation process. Therefore, a novel imaging technique has emerged, viscoelastography, which assesses both tissue elasticity and viscosity, the former linked to shear 39 wave speed and the latter linked to shear wave dispersion [2, 3].
3. Useful questions to at least address, if not answer include:
a. What are the advantages/disadvantages to use of large versus small region on interest measurements? Figure 2 appears to use a 1-2 mm ROI – why not use 3-5 mm?
b. What subsites within a gland are best to evaluate? Mention of subsites within the parotid may include the inability to evaluate the portion shielded by the mandible (“easily accessible” to ultrasound does not apply to this region). Most clinicians understand the general terms “facial process”, and “tail” to define parotid subsites, but an acknowledgement of the shortcomings to the less-than-uniform terminology addressing subsites compromises the capacity for shear-wave elastography to point out differences within a gland. The ‘gestalt’ grading systems used in ultrasound analysis of Sjogrens generally employ review of video clips encompassing imaging of much of the gland – the advantage of shear wave elastography (non-subjective quantitative assessment) is also a disadvantage in that only specific regions of interest within a gland are sampled – and warrant attention to the likelihood of differences in subunits (the tail of the parotid is likely to be altered than the facial process in the presence of a proximal obstructing stricture. An example of the vague reference to subsites in citation (30): Elasticity values in kiloPascal’s (kPa) were collected; the minimum, maximum,and mean values at 3 measurement points in different parts of the glands were taken
As identified by (30): “The number and size of regions of interests need also to be standardized.
4. Elastography characteristics changing over time: the cited support indicating value of elastography to characterize external beam induced radiation change to the glands is limited by short-term follow-up. For example, Kaluzny et al:
“The impact of time since RT for elasticity values is unclear. Directly after RT, in 6, 12, and 24 months the mean elastography values were 34.2 kPa, 36.6 kPa, 46.0 kPa, and 39.7 kPa, respectively, with differences that were not significant.”
These investigators further acknowledge the impact of time as changes to the gland evolve: The biggest difference in the mean elastography values was directly after RT (34.2 kPa) and after 12 months (46 kPa); in the further follow-up (24months) the value became lower (40 kPa), but this trend needs further examination
5. I would take issue with the concluding statement on p 6: “Therefore, SWE serves as a suitable method for identifying primary Sjögren's syndrome” The article cited (25) evaluated 15 articles to conclude that USE (ultrasound elastography) “demonstrates a high accuracy in discriminating between pSS and healthy/disease control groups”. By my interpretation, the SWE does not identify primary Sjogrens – as the authors also conclude: “Finally, this study specifically focuses on distinguishing pSS from healthy control groups, sicca symptoms, and secondary SS, but it does not evaluate the performance of USE in distinguishing pSS from other types of salivary gland diseases”
5. 6. The term “budging” – may have been intended to be “bulging” Budging tumors cause difficulties in applying a linear transducer
Comments on the Quality of English Language
slight revision suggested
Reviewer 2 Report
Comments and Suggestions for Authors
The authors present a summary of current research in the area of ​​salivary gland elastography. They assess the accuracy of different elastography methods in individual salivary gland pathologies. They indicate the methodological limitations of previously published studies examining elastography in salivary glands. Furthermore, they describe a new method of viscoelastography that can provide additional information in differential diagnosis. The article is an interesting and objective summary of current results of the accuracy of the elastography method in the assessment of salivary glands.
Reviewer 3 Report
Comments and Suggestions for Authors
Page 5 Line 184 In my opinion, the wording could be improved. Research about elastography in patients with Sjögren’s syndrome does not gained increased interest recently. The research in this field is ongoing for about 10 years. The authors even cited many of these first publication.
Page 7 Line 312 Typo in cytology
